# Oleic Acid Exhibits Anti-Proliferative and Anti-Invasive Activities via the PTEN/AKT/mTOR Pathway in Endometrial Cancer

**DOI:** 10.3390/cancers15225407

**Published:** 2023-11-14

**Authors:** Boer Deng, Weimin Kong, Hongyan Suo, Xiaochang Shen, Meredith A. Newton, Wesley C. Burkett, Ziyi Zhao, Catherine John, Wenchuan Sun, Xin Zhang, Yali Fan, Tianran Hao, Chunxiao Zhou, Victoria L. Bae-Jump

**Affiliations:** 1Department of Gynecology, Beijing Obstetrics and Gynecology Hospital, Capital Medical University, Beijing Maternal and Child Health Care Hospital, Beijing 100006, China; brdeng27@email.unc.edu (B.D.); hongyan@email.unc.edu (H.S.); xcshen96@email.unc.edu (X.S.); zhaoziyi@email.unc.edu (Z.Z.); xixi825@email.unc.edu (X.Z.); yaliy@email.unc.edu (Y.F.); 2Division of Gynecologic Oncology, University of North Carolina at Chapel Hill, Chapel Hill, NC 27599, USA; kwm1967@ccmu.edu.cn (W.K.); meredith.newton@unchealth.unc.edu (M.A.N.); wesley.burkett@unchealth.unc.edu (W.C.B.); catherine.john@unchealth.unc.edu (C.J.); wenchaun_sun@med.unc.edu (W.S.); tihao@med.unc.edu (T.H.); 3Lineberger Comprehensive Cancer Center, University of North Carolina at Chapel Hill, Chapel Hill, NC 27599, USA

**Keywords:** endometrial cancer, oleic acid, cell proliferation, invasion, PTEN/AKT pathway, lipid droplets

## Abstract

**Simple Summary:**

Metabolic reprogramming, especially fatty acid metabolism, is very strongly associated with carcinogenesis and progression of endometrial cancer. Increasing evidence revealed that oleic acid exhibited pro- and anti-tumor effects in different types of tumors, yet the mechanism underlying this dual effect on tumor growth remains unknown. In this study, we found that oleic acid significantly inhibited cell proliferation and tumor growth in endometrial cancer cells and a transgenic mouse model of endometrial cancer. Oleic acid increased lipogenesis and lipid droplets, and inhibition of the formation of lipid droplets resulted in an enhanced effect of oleic acid on inhibiting endometrial cancer cells. Importantly, oleic acid enhanced the expression of wild-type PTEN, while knockdown of PTEN or targeting AKT by ipatasertib partially reversed or amplified the effect oleic acid on cell proliferation and formation of lipid droplets, respectively. Thus, our results confirm that oleic acid inhibits cell growth depending on PTEN/AKT/mTOR pathway in endometrial cancer.

**Abstract:**

Reprogramming of fatty acid metabolism promotes cell growth and metastasis through a variety of processes that stimulate signaling molecules, energy storage, and membrane biosynthesis in endometrial cancer. Oleic acid is one of the most important monounsaturated fatty acids in the human body, which appears to have both pro- and anti-tumorigenic activities in various pre-clinical models. In this study, we evaluated the potential anti-tumor effects of oleic acid in endometrial cancer cells and the *LKB1^fl/fl^p53^fl/fl^* mouse model of endometrial cancer. Oleic acid increased lipogenesis, inhibited cell proliferation, caused cell cycle G1 arrest, induced cellular stress and apoptosis, and suppressed invasion in endometrial cancer cells. Targeting of diacylglycerol acyltransferases 1 and 2 effectively increased the cytotoxicity of oleic acid. Moreover, oleic acid significantly increased the expression of wild-type PTEN, and knockdown of PTEN by shRNA partially reversed the anti-proliferative and anti-invasive effects of oleic acid. Inhibition of the AKT/mTOR pathway by ipatasertib effectively increased the anti-tumor activity of oleic acid in endometrial cancer cells. Oleic acid treatment (10 mg/kg, daily, oral) for four weeks significantly inhibited tumor growth by 52.1% in the *LKB1^fl/fl^p53^fl/fl^* mice. Our findings demonstrated that oleic acid exhibited anti-tumorigenic activities, dependent on the PTEN/AKT/mTOR signaling pathway, in endometrial cancer.

## 1. Introduction

Endometrial cancer (EC) remains the most common gynecologic malignancy, with 65,950 new diagnoses and 12,550 deaths projected in the U.S. in 2023 [1]. Recognized endogenous and exogenous risk factors for the carcinogenesis of EC include increasing age, obesity and physical inactivity, early menarche and late menopause, low parity or infertility, family history, diabetes mellitus, unopposed estrogen exposure, tamoxifen use, genetic predisposition, and dietary factors [2,3]. Of these factors, obesity has been recognized as the risk factor most closely related to developing EC [4,5]. With the rising prevalence of obesity, EC is the only gynecologic malignancy with associated increased incidence [6,7,8,9]. The most common genetic alterations in EC are PTEN mutations or deletion and aberrational activation of the AKT/mTOR pathway due to loss of PTEN function and PI3K/AKT mutations, which occur in 60–80% of EC patients [10,11,12,13,14]. Early symptoms of postmenopausal vaginal bleeding ensure that most cases of EC are confined to the uterus and in a treatable stage at the time of diagnosis, with the prognosis for these patients being favorable after surgery (hysterectomy) +/− adjuvant radiation therapy [15]. However, patients with recurrent or more advanced stages do not respond well to conventional chemotherapy and lack effective treatment strategies, resulting in a poor prognosis [16]. Thus, focusing on the role of obesity-induced metabolic abnormalities in cell proliferation and tumor growth will help in better understanding the biological behavior of EC, and ultimately may provide insight on the potential manipulation of obesity-related metabolic pathways as novel treatment strategies for EC patients.

Lipid reprogramming, including fatty acid uptake, fatty acid biosynthesis, fatty acid oxidation, and fatty acid modification, is indispensable to meet the growth and survival demands of cancer cells by providing energy sources, altering biomembrane composition, and modulating signaling pathways [17,18]. Dietary lipids, as the primary source of exogenous fatty acids (FAs), effectively modify the diversity of the intracellular lipid pool and change the ratio of saturated FAs (SFAs) and unsaturated FAs (UFAs) on the cell membrane, thereby affecting metabolism and promoting or reducing tumor growth in a manner that depends on the amount and type of lipids [19,20,21,22]. Oleic acid (OA), which accounts for 4983% of all FAs in olive oil, is one of the most important monounsaturated fatty acids (MUFAs) in the human body [23]. OA has been found to have a significant effect on improvements in insulin sensitivity and liver secretory function, protecting against major chronic inflammatory diseases and reducing cardiovascular diseases [24,25,26]. Although some epidemiological studies have shown positive protective effects of olive oil against certain types of cancer, recent experimental studies have been controversial regarding the anti-tumorigenic activity of olive oil or OA. Treatment of the colon adenocarcinoma cell lines Caco-2 and HT-29 with olive oil or OA significantly induced cellular apoptosis and differentiation via downregulation of the expression of Cox-2 and Bcl-2 [27]. Long-term feeding with an OA-enriched diet reduced tumor diameter, increased tumor latency, and improved survival in a murine model of lung adenocarcinoma without affecting the incidence of pulmonary metastasis when compared with the mice fed a standard diet [21,28]. By contrast, OA at a dose of 30 µM increased cell proliferation and activated glycolysis through enhanced glucose transporter (GLUT) expression and activated peroxisome proliferator activated receptor (PPARa) in ovarian cancer cells [29]. Supplementation of OA in cervical cancer cells effectively stimulated cell proliferation, migration, and invasion through the FA transporter CD36 in a dose-dependent manner, and a high olive oil diet promoted tumor growth and metastasis in a mouse xenograft model of cervical cancer, indicating that increased intracellular OA concentrations and altered lipid pool composition may contribute to these OA-induced effects [30]. Despite multiple hypotheses for these conflicting results, there is currently no convincing and accepted explanation for the differential roles of OA in cancer cells.

Considering that obesity is the most important risk factor for the tumorigenesis and progression of EC, and that OA has diverse effects on cancer cell growth in different types of cancer, the investigation of the effects of OA on cell proliferation, tumor growth, and the PTEN/AKT pathway in EC will establish the basis for a more precise understanding of the biological impact of OA on EC. In this study, our aim was to evaluate the impact of OA on cell proliferation, cellular stress, apoptosis, cell cycle, invasion, formation of lipid droplets, and tumor growth in EC cell lines and a transgenic mouse model of EC, and to investigate the mechanisms underlying the role of OA on EC cell growth.

## 2. Materials and Methods

### 2.1. Cell Culture and Reagents

The human EC cell lines KLE and Hec-1B with wild-type PTEN expression were used in this study. The KLE cells were maintained in DMEM/F12 with 10% fetal bovine serum (FBS). The Hec-1B cells were cultured in McCoy’s 5A with 10% FBS. During OA treatment, the cell lines were cultured in media with 1% characterized fetal bovine serum (FBS; Thermo Scientific, Waltham, MA, USA). All media were supplemented with 100 U/mL of penicillin and 100 µg/mL of streptomycin. The cells were cultured in humidified 5% CO_2_ at 37 °C. OA and was purchased from Sigma-Aldrich (St. Louis, MO, USA). Bovine serum albumin (BSA; fatty acid free) (Sigma-Aldrich) was used to be conjugated with OA, at a 1:6.7 molar ratio (BSA/OA). To eliminate the effect of BSA on cell growth, the amount of BSA was kept constant in all treatment groups. Ipatasertib (IPAT) was purchased from MedChemExpress (Monmouth Junction, NJ, USA). Antibodies were purchased from Cell Signaling Technology (Beverly, MA, USA) and ABclonal (Woburn, MA, USA).

### 2.2. MTT Assay

Cell proliferation was assessed by MTT assay in the KLE and Hec-1B. Briefly, cells (4000–6000/well) were seeded in 96-well plates and cultured at 37 °C in 5% CO_2_ overnight and then treated with 0.1, 1, 10, 50, 100, 250, and 500 µM OA for 72 h. A total of 5 µL of MTT solution (5 mg/mL) was added to each well and incubated for another hour. The supernatants were aspirated, and 100 µL of DMSO was added to each well to lyse the formazan crystal. Optical density was measured at a wavelength of 570 nm with a Tecan microplate reader (Morrisville, NC, USA). The effect of OA on proliferation was assessed as a percentage of control cells in the same 96-well plates. IC50 values for KLE and Hec-1B cells were calculated by IC50 calculator (AAT Bioquest, Sunnyvale, CA, USA).

### 2.3. Colony Assay

The KLE and Hec-1B cells were plated in 6-well plates at a density of 400 cells/well overnight and exposed to the indicated concentrations of OA for 48 h. Cells were washed with PBS, and fresh media were added to each well, with the media being changed every 3 days for 12–14 days. At the end of the experiment, cells were fixed in 100% methanol for 15 min and stained with 0.5% crystal violet for 10 min. Colonies with over 50 cells were captured and counted under a Thermo Scientific Invitrogen EVOS microscope.

### 2.4. Cleaved Caspase 3, 8, 9 ELISA Assays

The KLE and Hec-1B cell lines were cultured in 6-well plates overnight and treated with different concentrations of OA (1, 50, 200 µM) for 14 h at 37 °C. After replacing the media with 1X caspase lysis buffer, cell lysates were collected and incubated with a reaction buffer containing 200 µM of caspase 3, 8, and 9 substrates (AAT Bioquest, Sunnyvale, CA, USA) for 30 min. Fluorescence was measured at Ex/Em = 400/505 nm for caspase 3, Ex/Em = 376/482 nm for caspase 8, and Ex/Em = 341/441 nm for caspase 9, using a Tecan plate reader. Experiments were repeated three times in triplicate.

### 2.5. Analysis of Cell Cycle by Cellometer

The KLE and Hec-1B cells were cultured in 6-well plates for 24 h and treated with OA (1, 50, 200 µM) for 36 h. The cells were harvested by 0.25% Trypsin (Sigma-Aldrich, St. Louis, MO, USA) and fixed in a 90% methanol solution for 1 h. The cells were resuspended in RNase A solution for 30 min, followed by incubation with propidium iodide (PI) staining solution for 10 min. All samples were immediately measured by Cellometer (Nexcelom, Lawrence, MA, USA) to assess cell cycle progression. The results were analyzed by FCS4 Express software (Molecular Devices, Sunnyvale, CA, USA).

### 2.6. Reactive Oxygen Species (ROS) Assay

The oxidation sensitive probe 2′,7′-dichlorofluorescin diacetate (DCFH-DA) assay was used to quantify intracellular ROS generation. The KLE and Hec-1B cells (8000–12,000 cells/well) were seeded in 96-well plates overnight and then treated with varying concentrations of OA (1, 50, and 200 µM) for 12 h. A total of 20 μL of 10 μM DCFH-DA was added to each well and incubated for 20 min at 37 °C in the dark. Fluorescence intensity was recorded at Ex/Em = 485/526 nm with a Tecan plate reader. Then, 10 µL of MTT solution (5 mg/mL) was added to each well using the same plate. MTT results were used to normalize fluorescence measurements on the same plate.

### 2.7. JC-1 Assay

The mitochondrial membrane potential was analyzed in KLE and Hec-1B cells using the fluorescent cationic dye JC-1 (AAT Bioquest, Sunnyvale, CA, USA). Cells were seeded in a 96-well plate overnight and then treated with 1, 50, and 200 μM OA for 6 h. A total of 1 µL of JC-1 (200 µM) was added to each well and incubated in the plate for 30 min at 37 °C in the dark. The levels of JC-1 monomers were detected at Ex/Em = 535/590 nm with a Tecan plate reader. Experiments were performed in triplicate and repeated three times to assess consistency.

### 2.8. TMRE Assay

The KLE and Hec-1B cells were seeded at a density of 2 × 10^5^ in black 96-well microplates overnight. The cells were treated with OA at 1, 50, and 200 µM for 10 h and then incubated with 1000 µM TMRE (tetramethylrhodamine ethyl ester; AAT Bioquest, Sunnyvale, CA, USA) for 30 min at 37 °C. After washing the plates with PBS, the plates were measured using a Tecan plate reader at Ex/Em = 549/575 nm.

### 2.9. Adhesion Assay

A total of 2.5 × 10^3^/well of KLE and Hec-1B cells were added in laminin-1-coated 96-well plates and then treated with 1, 50, and 200 μM OA for 1.5 hat 37 °C. The cells were fixed by 5% glutaraldehyde. Adhered cells were stained with crystal violet for 15 min, and 10% acetic acid was used to solubilize the dye. The absorbance was measured at 570 nm using a Tecan microplate reader.

### 2.10. Wound Healing Assay

The KLE and Hec-1B cells were plated at 3 × 10^5^ cells per well in 6-well plates for 24 h and then replaced with media with 0.5% charcoal stripped FBS for 12 h. A uniform wound was created through the cell monolayer using a 200 μL pipette tip. After washing three times with PBS in each well, cells were treated with OA for 48 h. Photographs were taken at 0, 24, and 48 h after scratching, and the cell migration capacity was analyzed by ImageJ software (National Institutes of Health, Bethesda, MD, USA).

### 2.11. Western Immunoblotting

The KLE and Hec-1B cells (2–3 × 10^5^ cells/well) were plated in 6-well plates overnight and then treated with indicated doses of OA 12 to 36 h. Cells were disrupted in ice-cold RIPA lysis buffer, and the supernatant was clarified by centrifugation at 12,000× *g* rpm for 20 min. Protein concentration was determined by a BCA Protein Assay kit (Thermo Fisher Scientific). Protein extracts were separated by SDS-polyacrylamide gel electrophoresis and transferred to an equilibrated polyvinylidene difluoride (PVDF) membrane at 4 °C. The membranes were blocked with 5% fat-free milk for 1 h at room temperature and incubated with the indicated primary antibodies overnight at 4 °C. After washing three times with TBS-T, the membranes were blocked with 5% fat-free milk with the indicated second antibodies (Cytiva, Marlborough, MA, USA) for 1 h. Target protein bands were visualized and quantified using the Western Lightning Plus-ECL (PerkinElmer, Waltham, MA, USA) on the ChemiDoc Image System (Bio-Rad, Hercules, CA, USA). Experiments were performed in duplicate to assess for consistency.

### 2.12. shRNA Transduction

PTEN shRNA lentiviral transduction plasmid (TRCN0000002747) and non-targeting shRNA lentiviral transduction plasmid (pLKO.1-puro Non-Target Control [SHC016V]) were obtained from Sigma-Aldrich; 293 T cells were used to package plasmids following the manufacturer’s protocol. After 72 h, supernatants were collected and passed through 0.45 μm filters to collect viruses. Hec-1B cells were transfected with PTEN shRNA or control virus for 72 h with 10 μg/mL polybrene and incubated with 2 μg/mL puromycin to screen transfected cells. Media were replaced every third day with fresh puromycin-containing media until stable clones were identified. PTEN knockdown was confirmed using Western immunoblotting analysis.

### 2.13. LKB1^fl/fl^p53^fl/fl^ Transgenic Mouse Model of EC

The *LKB1^fl/fl^p53^fl/fl^* genetically engineered mouse model of endometrioid EC was used in this study as detailed previously [31]. All mice were handled according to protocols approved by the Institutional Animal Care and Use Committee (IACUC) of the University of North Carolina at Chapel Hill (UNC-CH). Intrauterine Ad-Cre injections (5 μL recombinant Ad5-CMV-Cre [2.5 × 10^10^ P.F.U], Transfer Vector Core, University of Iowa) of *LKB1^fl/fl^p53^fl/fl^* mice were performed on the left uterine horn at six to eight weeks of age, to induce EC. The mice were further divided into the vehicle (same amount of BSA solution as OA group) or OA (10 mg/kg, daily, oral gavage, five days/week for four weeks) treatment groups at nine weeks of injection. Each group included 13 mice. The animals were weighed weekly throughout the study. All mice were euthanized after four weeks of OA or vehicle treatment. At sacrifice, endometrial tumors were weighed, and blood, liver, and intra-abdominal adipose tissue samples were taken. One-half of the endometrial tumors were snap-frozen and stored at −80 °C; the remaining one-half were fixed in 10% neutral-buffered formalin and paraffin-embedded.

### 2.14. Immunohistochemistry (IHC) of Endometrial Tumors

The mouse endometrial tumor tissues were formalin-fixed and paraffin-embedded at the Animal Histopathology Core Facility at UNC-CH. The slides (4 μm) were first deparaffinized and hydrated with the following buffer: xylene for 5 min twice, 100% ethanol for 5 min twice, 95% ethanol for 5 min twice, 85% ethanol for 5 min, 70% ethanol for 5 min, 50% ethanol for 5 min, 30% ethanol for 5 min, and DD water for 5 min twice. Subsequently, an appropriate antigen retrieval buffer was added to the slides and boiled for 20 min, followed by soaking in cold water for 10 min. The slides were incubated with protein block solution (Dako, Agilent Technologies, Santa Clara, CA, USA) for 1 h and then with the primary antibodies for Ki-67 (1:400), p-ACC (acetyl-CoA carboxylase, 1:800), VEGF (Vascular endothelial growth factor, 1:800), ATGL (adipose triglyceride lipase, 1:100), and Bcl-xL (B-cell lymphoma-extra-large, 1:1200) for 2 h at room temperature. The slides were washed three times with TBS-T washing buffer and incubated with appropriate secondary antibodies (Biotinylated goat anti-rabbit, Vector Labs, Burlingame, CA, USA) at room temperature for 1 h. Further processing was carried out using ABC-Staining Kits (Vector Labs). IHC slides were scanned by Motic (Houston, TX, USA) and scored by ImagePro software (Vista, CA, USA).

### 2.15. Hematoxylin-Eosin (HE) Staining

The formalin-fixed and paraffin-embedded (FFPE) sections were dipped in hematoxylin for 5 min, covered with 0.1% HCl and 0.1% NH_4_OH, and washed with running tap water. Subsequently, the sections were stained with eosin. The stained sections were scanned by Motic (Houston, TX, USA) and scored by ImagePro software (Vista, CA, USA).

### 2.16. Statistical Analysis

All data were presented as the mean ± standard deviation. The differences between two groups were analyzed using unpaired Student’s *t*-test. One-way analysis of variance (ANOVA) with Tukey’s multiple comparison test was employed to perform statistical comparison of multiple groups. Two-way ANOVA was used to analyze the differences between non-transfected and transfected PTEN Hec-1B cells. GraphPad Prism 8 (La Jolla, CA, USA) was used for comparisons, and all graphs with *p* values of <0.05 were considered to have significant group differences.

## 3. Results

OA inhibited cell proliferation and tumor growth in EC cell lines and the *LKB1^fl/fl^p53^fl/fl^* transgenic mouse model of EC.

The EC cell lines, KLE, Hec-1B, ECC-1, Ishikawa, and AN3CA, were used to examine the effect of OA on cell proliferation. Cells were exposed to OA at varying concentrations from 0.1 to 500 μM for 72 h. The results of the MTT assay demonstrated that with increasing doses of OA, cell viability decreased in a dose-dependent manner in all of the EC cell lines. The mean IC50 values of OA were 445.6 μM for KLE cells, 382.8 μM for Hec-1B cells, 369.8 μM for ECC-1 cells, 6762 μM for AN3CA cells, and 2219 μM for Ishikawa cells (Figure 1A). To observe the effect of OA on cell proliferation with different treatment times, the KLE and Hec-1B cells were treated with 50 μM and 200 μM OA, respectively, for 24, 48, and 72 h. The MTT results showed that OA significantly inhibited cell proliferation in a time-dependent manner (Figure 1B). After treatment of both cell lines with OA at doses of 50 and 200 μM for 48 and 72 h, the effects of cell proliferation at 200 μM OA produced more potent inhibitory activity compared with 50 μM OA in the KLE and Hec-1B cells (Figure 1B). Since the colony assay is an effective method to measure the long-term effects of cytotoxic agents on cell proliferation ability, the KLE and Hec-1B cells were treated with 1, 50, and 200 μM OA for 48 h, and then the cells were cultured for 12 additional days. OA at a dose of 200 μM significantly inhibited the colony formation of KLE cells and Hec-1B cells with 70.6% and 68.5% inhibition compared to control cells, respectively (Figure 1C). These results confirm that OA inhibited cell proliferation of EC cells in vitro.

To assess the effect of OA on tumor growth in vivo, the *LKB1^fl/fl^p53^fl/fl^* transgenic mouse model of EC was treated with OA at 10 mg/kg daily by oral gavage for four weeks (13 mice/group). OA effectively decreased tumor weight compared to the vehicle group (0.39 g vs. 0.79 g, *p* < 0.01) after four weeks of treatment (Figure 1D). IHC staining showed that OA reduced the expression of Ki-67 by 23.3% in EC tumors compared with control mice (*p* = 0.018, Figure 1E). In addition, during the treatment period, the mice were active and did not gain or lose body weight. Given that gonadal fat pad weight mainly increases during the initial phase of weight gain, the gonadal fat pad and liver were weighed. Gonadal fat pad weight and liver weight did not differ between OA-treated and control mice. However, hematoxylin and eosin (H&E) staining results demonstrated that gonadal adipocytes were larger, and the hepatocytes contained large lipid droplets in the OA-treated group (Figure 1F), suggesting that four weeks of OA treatment increased lipid synthesis in adipose tissue and liver, although OA did not increase body weight.

OA induced cellular stress, autophagy, and cell apoptosis in EC cells.

To investigate the role of OA in cellular stress, the KLE and Hec-1B cells were treated with 1, 50, and 200 μM OA for 6 h. The ROS assay showed that OA at doses of 50 and 200 μM significantly increased cellular ROS production in both cell lines. OA at 200 μM increased the level of ROS by 1.23 times in the KLE cells and by 1.329 times in the Hec-1B cells, compared to untreated cells (Figure 2A). To further characterize the effect of OA on mitochondrial membrane potential, TMRE and JC-1 assays were employed to detect alterations of mitochondrial membrane potential after 8 h of OA exposure. As expected, 50 and 200 μM OA reduced mitochondrial membrane potential in both cell lines. OA at 200 μM significantly decreased TMRE levels in the KLE and Hec-1B cells by 29.1% and 10.1%, respectively, compared to control cells (Figure 2B). Similarly, treatment of OA at a dose of 200 μM for 6 h reduced JC-1 levels in KLE and Hec-1B cells by 22.5% and 22.3%, respectively, compared to untreated cells (Figure 2C). Western immunoblotting results showed that OA increased expression of the cellular stress-related proteins Bip, protein kinase R-like endoplasmic reticulum kinase (PERK), and Erol-1, and increased the expression of the autophagy-related proteins Beclin-1 and Atg12 after 24 h of treatment in the KLE and Hec-1B cell lines (Figure 2D and Appendix A).

Given that dysregulation of ROS and autophagy triggers apoptotic signaling pathways, the levels of cleaved caspase 3, 8, and 9 were detected by ELISA. Similar to the ROS results, OA at doses of 50 and 200 μM significantly increased the levels of cleaved caspase 3, 8, and 9 in the KLE and Hec-1B cells after 12 h of treatment. OA at 200 μM increased the level of cleaved caspase 3, 8, and 9 by 1.37-fold, 1.27-fold, and 1.36-fold in the KLE cells, respectively, and increased the levels of cleaved caspase 3, 8, and 9 by 1.44-fold, 1.37-fold, and 1.48-fold in the Hec-1B cells, respectively, compared with untreated cells (Figure 2E). Meanwhile, Western immunoblotting showed that treatment of cells with OA for 8 h decreased the expression of Bcl-xL and MCL-1 in both cells (Figure 2F and Appendix A). IHC results demonstrated that OA reduced the expression of Bcl-xL by 21.7% in OA-treated EC tumors from *LKB1^fl/fl^p53^fl/fl^* mice compared with control mice (Figure 2G). Additionally, decreased expression of Bcl-xL was observed in the mouse EC tumors detected by Western immunoblotting (Figure 2H and Appendix A). These results indicate that OA induces cell apoptosis in EC cells in vitro and in vivo.

OA induced G0/G1 phase arrest in EC cells.

To investigate whether OA impacts cell cycle progression of EC cells, we assessed the cell cycle profile of the KLE and Hec-1B cells after 36 h of OA treatment using a Cellometer. Results showed that OA increased the G0/G1 phase and decreased the G2 phase of both cell lines in a dose-dependent manner. OA at a dose of 200 μM increased the G0/G1 phase from 56.86% in the untreated group to 67.60% in the KLE cells, and increased G0/G1 phase from 36.83% to 51.87% in Hec-1B cells (Figure 3A). Furthermore, Western immunoblotting results indicated that OA decreased the expression level of CDK4 and CDK6 in both cell lines after 24 h of treatment (Figure 3B and Appendix A). These results confirmed that OA induced G0/G1 phase arrest in EC cells.

OA inhibited adhesion and invasion in EC.

To investigate the effects of OA on cell adhesion and migration, we employed the laminin-1 adhesion and wound healing assays in the KLE and Hec-1B cells. Additionally, 50 and 200 μM OA significantly inhibited the adhesion of KLE and Hec-1B cells. OA at a dose of 200 μM inhibited adhesion by 8.5% and 13.3% in KLE and Hec-1B cells, respectively, compared to untreated cells (Figure 4A). Similarly, the wound healing assay showed that OA reduced the migratory ability in both cell lines after 28 h of treatment. OA at a dose of 200 μM significantly increased wound healing width by 1.66-fold and 1.51-fold in the KLE and Hec-1B cells, respectively, as compared to the untreated cells (Figure 4B). Given that epithelial mesenchymal transition (EMT) is an essential step for cell invasion and metastasis in cancer, EMT-related proteins were detected by Western immunoblotting after treatment of the KLE and Hec-1B cells with OA for 24 h. The results showed that OA decreased the expression of N-cadherin and Snail and increased the level of Slug in both cells (Figure 4C and Appendix A). Due to the role of angiogenesis in the process of cell invasion, the expression of vascular endothelial growth factor (VEGF) in mouse EC tissues was detected by IHC. OA significantly reduced the expression of VEGF by 24.9% in OA-treated tumor tissues compared to control mice (Figure 4D). These results confirmed that OA effectively suppressed cell adhesion and invasion migration in EC.

OA induced the accumulation of intracellular lipid droplets in EC cells.

Since OA increased the size of gonadal adipocytes and promoted lipid droplet (LD) formation in hepatocytes in *LKB1^fl/fl^p53^fl/fl^* mice, we hypothesized that OA might increase lipid metabolism in EC cells and tumors. Oil red O staining was used to detect the content of LDs in KLE and Hec-1B cells. After 24 h of treatment, OA at doses of 50 and 200 μM significantly increased accumulation of LDs in both cell lines. A total of 200 μM OA increased LD content by 2.27-fold and 1.93-fold in the KLE and Hec-1B cells, respectively, compared to untreated cells (Figure 5A). Western blotting revealed that OA increased the expression of FAS, and the effect was more effective at a dose of 50 μM. In addition, increased expression of DGAT1 and DGAT2 and decreased expression of CPT1A and Glut1 were observed in both cell lines after 24 h of treatment. In KLE cells, OA at a dose of 200 μM also reduced the expression of ATGL, glucose transporter type 4 (Glut 4), and lactate dehydrogenase A (LDHA), while in Hec-1B cells, OA increased the expression of Glut4 and LDHA (Figure 5B and Appendix A). Similarly, IHC staining showed that OA increased p-ACC expression by 1.29-fold in OA-treated EC tumors compared with control mice (Figure 5C). These results suggested that OA activates lipogenesis and inhibits FA oxidation and glycolytic activity, and that the two EC cell lines responded differently to OA treatment.

Given that DGAT1 and DGAT2 catalyze the final step in the synthesis of triacylglycerol (TAG), we next investigated the role of each DGAT isoenzyme in the increased formation of LDs induced by OA treatment in EC cells. The KLE and Hec-1B cells were treated with specific inhibitors of DGAT1 (T-863) or DGAT2 (PF-06424439), or a combination of both inhibitors. T-863 or PF-06424439 were effective in reversing OA-induced DGAT1 or DGAT2 expression in both cells after 24 h of treatment (Figure 5D and Appendix A). MTT assay showed that 2.5 μM T-863 or 5 μM PF-06424439, or the combination of T-863 and PF-06424439, significantly inhibited cell proliferation, and that the combination of T-863 or PF-06424439 and 100 μM OA produced more potent inhibition of cell proliferation in both cell lines after 72 h of treatment. Importantly, the triple combination of T-863, PF-06424439, and OA showed the most potent inhibitory effect in both cell lines compared to single agent treatment or any combination of just two of these agents (Figure 5E). Next, the alterations of LD content were determined using Oil red O staining after 24 h of treatment with OA combined with and without T-863 and PF-06424439 in the KLE and Hec-1B cells. Neither T-863 (2.5 μM) nor PF-06424439 (5 μM) affected LD content when used as a single agent in non-OA-treated cells. The combination of T-863 and PF-06424439 significantly decreased the content of LDs in KLE cells, while showing no significant change in LD content in Hec-1B cells in non-OA-treated cells. Treatment of 100 μM OA for 24 h significantly increased the content of LDs by 1.77-fold in KLE cells and 1.48-fold in Hec-1B cells. After 24 h of OA treatment, 2.5 μM T-863 decreased LD content by 16.7% and 14.6% in the KLE and Hec-1B cells, respectively, while 5 μM PF-06424439 reduced LD formation by 11.0% and 10.9% in the KLE and Hec-1B cells, respectively. However, the combination of T-863 and PF-06424439 effectively reversed OA induced formation of LDs in both cell lines (Figure 5F). These results indicated that a high dose of OA increases lipogenesis and formation of LDs, and that inhibition of DGAT1 and DGAT2 activity reduces LD content and enhances the cytotoxicity of OA in EC cells.

PTEN regulates the formation of LDs in EC cells.

Given that the PTEN/AKT/mTOR and MAPKs pathways are vital signaling cascades that regulate metabolism and respond to stressful stimuli, we therefore speculated that these pathways might be involved in the growth inhibitory effect of OA on KLE and Hec-1B cells. Both cell lines were treated with OA at concentrations of 1, 50, and 200 μM for 24 h. Immunoblotting analysis was used to examine the effect of OA on the PTEN/AKT/mTOR and MAPKs pathways. The KLE and Hec-1B cells have been reported to have wild-type PTEN. OA significantly increased the expression of PTEN and decreased the expression of phosphorylated-PTEN, phosphorylated AKT (ser473), and phosphorylated S6 in both cells. In addition, OA induced the expression of phosphorylated-p42/44 in both cell lines, increased phosphorylation of p38 expression in the KLE cells, and decreased the expression of p38 phosphorylation in the Hec-1B cells (Figure 6A and Appendix A). In randomly paired *LKB1^fl/fl^p53^fl/fl^* endometrial tumors, Western blotting results showed that OA treatment for four weeks led to downregulation of phosphorylated-S6 expression (Figure 6B and Appendix A). These results indicate that the PTEN/AKT/mTOR and MAPKs pathways may play a role in the inhibitory effect of OA on cell proliferation in EC cells.

Considering that PTEN mutation and deletion are the most common molecular alterations in EC, and OA significantly increases the expression of PTEN in the KLE and Hec-1B cell lines, we performed shRNA-mediated knockdown of PTEN in the Hec-1B cells to examine the impact of PTEN loss on OA-induced cell growth. Transfection of shPTEN effectively reduced the expression of PTEN compared to the untreated control and shRNA lentiviral vector (shCtrl) control. Meanwhile, shPTEN increased phosphorylated-AKT and phosphorylated-S6 and decreased the expression of CDK4 and Bip compared to the control cells. MTT assay showed that loss of PTEN significantly increased cell proliferation in a time-dependent manner (Figure 6C and Appendix A). Next, we evaluated the inhibitory effects of OA on cell growth in the shPTEN cells. Downregulation of PTEN significantly increased the IC50 values of cells against OA after 72 h of treatment compared with shCtrl cells and non-transfected cells. After 72 and 96 h of culture and growth, the proliferative ability of shPTEN cells was significantly higher than that of non-transfected and shCtrl cells (Figure 6D). Similar results were found in the colony assay in the shCtrl and shPTEN cells. After 48 h of exposure to OA and a subsequent 12 days of cell culture growth, 200 μM OA inhibited colony formation by 49.7% in the shCtrl cells and 28.4% in the shPTEN cells compared to non-transfected cells (Figure 6E). JC-1 and caspase 3 assays showed that loss of PTEN partially restored mitochondrial membrane potential and cleaved caspase 3 activity in OA-treated shPTEN cells (Figure 6F). We next used Oil red O staining to examine the effect of the knockdown of PTEN on the formation of LDs. Non-transfected, shCtrl, and shPTEN cells were treated with 200 μM OA for 24 h, and the results showed that Oil red O content in the shPTEN cells decreased by 20.9% and 22.9% compared to the non-transfected and shCtrl cells, respectively (Figure 6G). To explore the underlying mechanisms by which loss of PTEN affect cell growth and LD formation, we performed Western immunoblotting to determine the effect of PTEN loss on the DGATs, apoptotic proteins, cellular stress proteins, and downstream targets of the AKT/mTOR pathways in the shPTEN cells. The results demonstrated that loss of PTEN decreased OA-induced expression of DGAT1 and DGAT 2 (Figure 6H and Appendix A), suggesting that knockdown of PTEN effectively reduced lipogenesis. Additionally, loss of PTEN partially reversed the stimulatory effects of OA on the expression of phosphorylated-AKT, phosphorylated-S6, Bip, Bcl-xL, Bcl-2, and CDK4 (Figure 6I and Appendix A).

Effects of targeting AKT on OA-induced cell growth and LD formation in EC cells.

Given the role of PTEN in OA-induced cell growth and LD formation, we next investigated whether PTEN/AKT/S6 signaling is involved in the inhibitory effect of OA on EC cells. IPAT, a potent AKT inhibitor, has been shown to inhibit the AKT/mTOR pathway in EC cells and is in multiple clinical trials in cancer patients. KLE and the Hec-1B cells were treated with 20 μM IPAT, 200 μM OA, and the combination of IPAT and OA for 72 h. MTT results showed that the combination treatment produced more potent inhibitory effects on cell proliferation compared with the treatment with single agent in both cells (Figure 7A). Colony assay also supported the results of the MTT assay (Figure 7B). IPAT significantly increased the level of cleaved caspase 3 in both cells and reduced mitochondrial membrane potential in Hec-1B cells, with the combination treatment producing the most pronounced effects (Figure 7C). The results of Western blotting showed that, compared to IPAT alone, OA alone, and control cells, the combination of IPAT and OA had a stronger inhibitory effect on S6 phosphorylation in both cell lines. Similar results were found for the effect of combination therapy on the expression of PDI, Bip, Bcl-xL, and Bax in both cells, where IPAT and OA combined increased Bax, PDI, and Bip expression and decreased the expression of Bcl-xL (Figure 7D and Appendix A).

To examine whether inhibition of the AKT/mTOR pathway by use of IPAT impacts the formation of LDs in the KLE and Hec-1B cells after treatment with OA, the KLE and Hec-1B cells were treated with an OA (200 μM), IPAT (20 μM), and a combination treatment for 24 h to evaluate LD formation. Results of Oil red O staining showed that the combination of OA and IPAT enhanced OA-induced LD formation in both cells (Figure 7E). To examine the effects of OA and IPAT on DGATs activity, the KLE and Hec-1B cells were treated with OA (200 μM), IPAT (20 μM), and OA combined with IPAT for 24 h. IPAT increased OA-induced expression of DGAT1 and DGAT2 in both cell lines (Figure 7F and Appendix A). Collectively, these results suggested that the effects of OA cell growth and LD formation partially depend on the PTEN/AKT/S6 pathway in EC cells.

OA-inhibited adhesion and invasion in EC cells.

To investigate the regulation effect of the PTEN/AKT/mTOR pathway on OA-induced adhesion and migration of EC cells, non-transfected, shCtrl, and shPTEN cells were treated with OA at a dose of 200 μM for 28 h, and wound healing was used to detect the migratory ability of these cells. The results showed that knockdown of PTEN significantly increased the migration of Hec-1B cells, and the inhibitory effect of OA on cell migration was attenuated after knockdown of PTEN (Figure 8A). Western blotting results showed that knockdown of PTEN decreased the expression of Slug and increased the expression of Snail compared to the non-transfected and shCtrl cells (Figure 8B and Appendix A). Loss of PTEN attenuated the effects of OA on the expression of Snail and Slug (Figure 8C and Appendix A).

To further confirm the role of the PTEN/AKT/mTOR pathway in regulating the effects of OA on cell migration, KLE and Hec-1B cells were treated with OA (200 μM), IPAT (20 μM), and OA combined with IPAT for 28 h. The results of wound healing showed that IPAT alone did not affect the migratory ability of either cell line, and the combination of IPAT and OA significantly increased the inhibitory effects of OA on cell migration in both cell lines compared to IPAT or OA alone treatment (Figure 8D). Western blotting results showed that treatment of KLE and Hec-1B cells with 200 μM OA reduced the expression of EMT-related protein Vimentin compared to untreated cells. The combination of IPAT and OA showed a more potent inhibitory effect on Vimentin in both cells (Figure 8E and Appendix A). Overall, these results indicated that OA inhibited migration ability of EC cells depending on PTEN/AKT/S6 pathways.

## 4. Discussion

Determining the effect of FAs on EC tumor growth and their mechanisms of action is critical to better understanding the role of obesity and abnormal lipid metabolism in EC. Although some epidemiological studies have shown that OA-rich diets are associated with protection against breast and pancreatic cancers, OA exhibits dual activities of inhibiting or promoting tumor cell proliferation and growth in pre-clinical models through multiple, inter-related cell signaling and metabolic pathways [32,33]. Treatment of OE19 and OE33 esophageal cancer cell lines with OA significantly inhibited cell proliferation through upregulation of p53, p21, and 27 expression, and activation of the AMP-activated protein kinase (AMPK) pathway [34]. Exogenous supplementation with OA-inhibited Her-2/neu overexpression and cell proliferation, and synergistically enhanced trastuzumab-induced inhibition of cell growth and apoptosis in breast cancer cells with Her-2/neu amplification [35]. By contrast, 30 µM OA promoted cell proliferation and enhanced glycolytic activity through activation of peroxisome proliferator activated receptor α (PPARα) and the BRAD4-L-MYC-GLUT axis in ovarian cancer cells [29]. Treatment of OA at a dose of 100 µM stimulated cell viability, increased GPR40 expression, and effectively sensitized breast cancer cells to OA. Despite obesity and abnormal metabolism being high-risk factors that promote the development of EC, which has been confirmed decades ago, the impact of OA on the growth of EC tumor cells has been rarely reported so far. In this study, using EC cell lines and the *LKB1^fl/fl^p53^fl/fl^* transgenic mouse EC model, we found that OA effectively inhibited cell proliferation, induced cell stress and apoptosis, caused cell cycle G1 arrest, reduced cell invasion and migration, and suppressed tumor growth. Supplementation with OA increased lipogenesis and LD formation, whereas targeting DGATs significantly reduced cell proliferation and LD accumulation in EC cells. Importantly, OA was able to increase the expression of wild-type PTEN and decreased phosphorylation of PTEN in a dose-dependent manner in the Hec-1B cells. Knockdown of PTEN effectively decreased lipogenesis and LD formation and reduced the sensitivity of EC cells to OA. These results confirmed that the anti-tumorigenic activity of OA is mediated at least in part by its dependency on the PTEN/AKT/mTOR pathway in EC.

The anti-proliferative effects of OA involve multiple mechanisms, including cell cycle arrest, cellular stress, autophagy, and apoptosis in cancer cells. Different types of FA promote cancer cell autophagy via different molecular mechanisms, and autophagy regulates FA’s availability via mitochondria-endoplasmic reticulum contact sites [36,37]. Emerging evidence for an interaction between core proteins in the cellular stress, autophagy, and apoptotic pathways underlies the molecular mechanism of crosstalk between cellular stress, apoptosis, and autophagy [38,39]. In addition, cell cycle arrest provides an opportunity for cancer cells to undergo repair mechanisms or follow apoptotic pathways [40]. Previous studies have documented that monounsaturated fatty acids (MUFAs) including OA-inhibited cell proliferation by G0/G1 phase arrest in tongue squamous cell carcinoma and HepG2 cells [22,41]. Inhibition of cell proliferation and induction of annexin V expression by OA in hepatocellular carcinoma cell lines is dependent on autophagic flux [42]. Treating YAC-1 lymphoma cells with OA increased ROS level and caspase 3 activity [43]. OA-induced apoptosis appears to involve multiple mechanisms, including the Cox-2, AMPK/S6, and STAT3 pathways, in colorectal, esophageal, and breast cancer cells [27,34,44]. Consistent with these results, our results showed that OA treatment at concentrations higher than 50 µM effectively increased intracellular ROS levels, decreased the levels of mitochondrial membrane potential, and upregulated the expression of Bip, PERK, Erol-1, Beclin-1, and Atg12 in EC cells. Meanwhile, OA at doses of 50 and 200 µM significantly increased cleaved caspase 3, 8, and 9 activities and decreased the expression of Bcl-xL and MCL-1, suggesting that intrinsic and extrinsic apoptotic pathways participate in OA-induced cell inhibition in EC cells. Thus, our data underline that OA-induced cellular stress and autophagy may be the triggers for apoptosis in EC cells.

Deletion or mutation of PTEN and genetic alterations in PI3K and AKT make PTEN/PI3K/AKT/mTOR signaling the most dysregulated signaling pathway controlling tumorigenesis, cell proliferation, apoptosis, invasion, and metabolism in EC. Deficiency of PTEN and activation of PI3K/AKT can increase glycolytic activity and regulate FA biosynthesis in cancer cells [45,46,47]. Treatment with increasing concentrations of PA resulted in decreased expression of PTEN in a dose-dependent manner through T366 phosphorylation and protein degradation of PTEN in U2OS cells and mouse livers [48]. Similarly, treating HepG2 cells with OA also reduced the expression of PTEN and promoted tumor growth in HepG2-derived tumors in nude mice fed an oleic acid-rich diet [49]. By contrast, our results show that treatment of KLE and Hec-1B cells expressing wild-type PTEN with 1, 50, and 200 µM OA for 24 h significantly increased the expression of PTEN and decreased the expression of phosphorylated-PTEN, AKT, and S6 in both cell lines. Knockdown of PTEN in Hec-1B cells attenuated the cytotoxicity of OA and decreased TG synthesis and LD formation. Since PTEN mainly targets and dephosphorylates PIP3, which acts as a major passive regulator of PI3K/AKT signaling, we used the AKT inhibitor IPAT to inhibit the AKT/mTOR pathway to observe the effect of OA on cell growth, thereby further verifying the role of PTEN in the inhibition of cell growth by OA. As we expected, IPAT effectively restored the effects of OA on cell proliferation, apoptosis, and LA formation in both cell lines. Our results confirm that the status of PTEN and the activity of the AKT/mTOR pathway are key factors controlling the effect of OA on cell proliferation in wild-type PTEN EC cells. Functional PTEN is tightly regulated in cancer cells at the transcriptional, post-transcriptional, and post-translational levels, highlighting the complexity of the mechanisms underlying the effects of OA on PTEN expression in different types of cancer cells [50]. Recent studies have proposed that pronounced imbalance of FAs, in particular the composition of phospholipids in cell membranes, has the capacity to regulate tumor cell growth and the activation of relevant signaling cascades [51,52]. Thus, our further work will investigate the relationship between EC membrane FA composition and functional PTEN.

Given the dual effects of OA on cell proliferation, it is not surprising that OA can also promote or inhibit cancer cell invasion and migration. Treatment of breast cancer cells MDA-MB-231 and MCF-7 with 200 or 400 µM OA promoted cell proliferation, increased the expression of MMP9, and enhanced invasive and migratory ability through activation of AMPK, epidermal growth factor receptor (EGFR), protein kinase C (PKC), AKT, and Src-dependent pathways [53,54,55]. In colorectal cancer cells, OA-induced cellular EGFR stress is critical for increased cell invasion, and treatment of anti-oxidants dramatically inhibited circulating OA-enhanced metastatic seeding of tumor cells in the lungs of mice [56]. On the other hand, several studies exhibited that OA effectively inhibited cell invasion and migration in esophageal cancer and hepatocellular carcinoma cells [34,42]. Additionally, OA decreased DNA synthesis and angiogenesis in Lewis lung carcinoma cells and inhibited the metastasis from spleen to the liver in Lewis lung carcinoma–bearing mice [21]. Similar to these other studies, we found that 50 and 200 µM OA inhibited cell adhesion and migration, while decreasing the expression of N-cadherin and Snail and increasing the expression of Slug in both cell lines. Importantly, treatment of *LKB1^fl/fl^p53^fl/fl^* mice with OA for four weeks significantly reduced the expression of VEGF in EC tumors compared to the untreated mice, suggesting that OA is capable of mediating processes responsible for invasion and migration in EC. To verify the role of the PTEN/AKT pathway in OA reducing invasion, we knocked down PTEN in Hec-1B cells and inhibited AKT activity with IPAT in Hec-1B and KLE cells and analyzed the changes in wound healing and EMT-related proteins. The results allow us to conclude that OA reduced invasion and migration is at least partially dependent on the PTEN/AKT/mTOR pathway since knockdown attenuated the ability of OA to reduce invasion and migration, whereas inhibition of the AKT pathway effectively increased the ability of OA to reduce invasion.

Metabolic disorders in obesity can lead to ectopic accumulation of LDs in non-adipose tissues. Increased synthesis and accumulation of LDs are essential for tumor cells to remove intracellular free FAs and provide sufficient ATP and biomolecules for the proliferation and invasion of tumor cells [57,58]. Exposure to OA significantly increased lipogenesis and the LD formation in cancer cells to dampen the potential damaging surge of lipids [59,60]. The metabolic balance of FA in cancer cells is closely related to the tight regulation between key enzymes of lipogenesis and lipolysis, such as ATGL, DGATs, and FAS, and the molecular mechanisms regulating FA metabolic balance in EC cells are largely unknown [61]. In this current study, we found that LD formation and the expression of DGAT1 and DGAT2 were stimulated with increasing OA concentrations in the KLE and Hec-1B cell lines. Treatment of *LKB1^fl/fl^p53^fl/fl^* mice with OA resulted in LD formation in the liver, reduced the expression of CPT1A, and increased phosphorylated ACC but not ATGL expression in the endometrial tumors, indicating that OA increases lipogenic but not lipolytic activity in EC. Combination treatment with OA and the DGAT1 and DGAT2 inhibitors significantly attenuated OA-induced LD accumulation and increased OA-induced anti-proliferative activity in both cell lines. Knockdown of PTEN reduced OA-induced LD formation and expression of DGAT1 and DGAT2, whereas inhibition of AKT by IPAT had opposite effects on LD formation and the expression of DGATs. Our findings suggest that loss of PTEN reduced lipogenesis and LD accumulation, leading to increased energy availability to sustain increased cell proliferation.

## 5. Conclusions

In conclusion, our results obtained in EC cells and the *LKB1^fl/fl^p53^fl/fl^* mouse model of EC provide evidence that OA inhibits cell proliferation, invasion, and tumor growth, and increases lipogenesis and LD formation. OA-induced PTEN expression also provides a novel mechanistic link between the PTEN/AKT pathway and OA-induced inhibition of growth and lipogenesis, indicating that anti-proliferative and anti-invasive effects of OA in EC are partially dependent on signaling through the PTEN/AKT/mTOR pathway. Although the role of OA in reducing EC cell growth and increasing functional PTEN is novel for this study, the mechanism by which OA regulates PTEN expression and function requires further exploration.

## Figures and Tables

**Figure 1 cancers-15-05407-f001:**
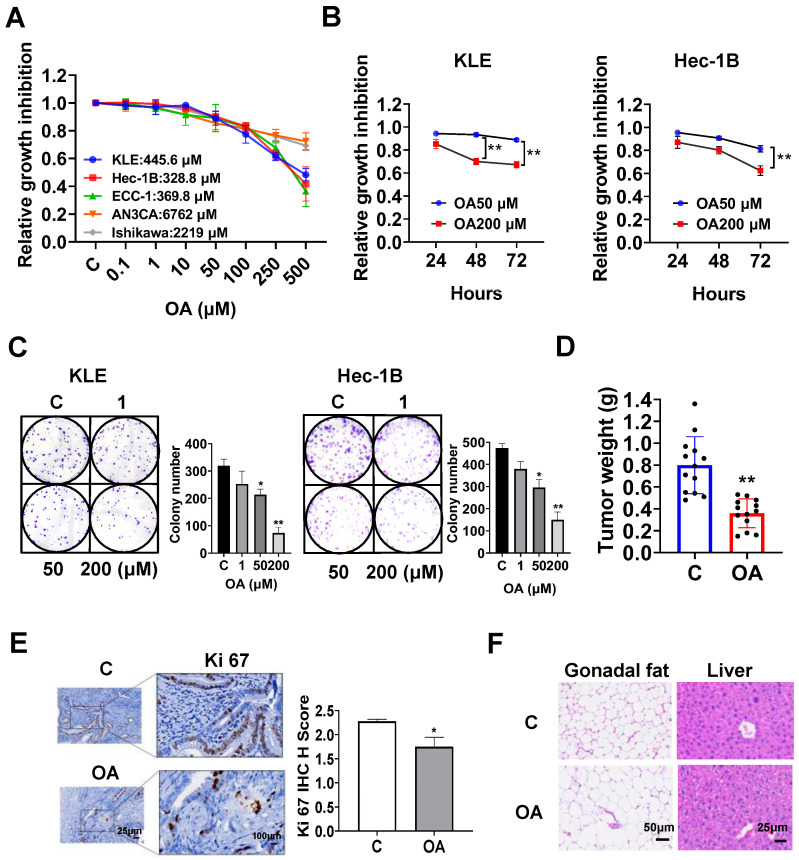
OA inhibited cell proliferation in EC cell lines and tumor growth in *LKB1^fl/fl^p53^fl/fl^* mice. The KLE, Hec-1B, ECC-1, Ishikawa, and AN3CA cells were treated with the indicated doses of OA for 72 h. Cell proliferation was detected using an MTT assay. OA inhibited cell growth in all four cell lines in a dose-dependent manner (**A**). OA inhibited cell growth in a time-dependent manner in KLE and Hec-1B cells (**B**). Treatment of OA at doses of 50 and 200 μM for 48 h significantly inhibited the formation of colony in KLE and Hec-1B cells (**C**). *LKB1^fl/fl^p53^fl/fl^* mice were treated with OA (10 mg/kg, oral, daily) or vehicle for four weeks, and the results showed that OA effectively reduced tumor weight compared with control mice (**D**). IHC results showed that OA treatment reduced the expression of Ki-67 in endometrial tumor tissues of *LKB1^fl/fl^p53^fl/fl^* mice (**E**). HE staining results showed that OA treatment increased the adipocyte size in adipose gonadal fat tissues and induced hepatocyte steatosis in *LKB1^fl/fl^p53^fl/fl^* mice (**F**). * *p* < 0.05, ** *p* < 0.01.

**Figure 2 cancers-15-05407-f002:**
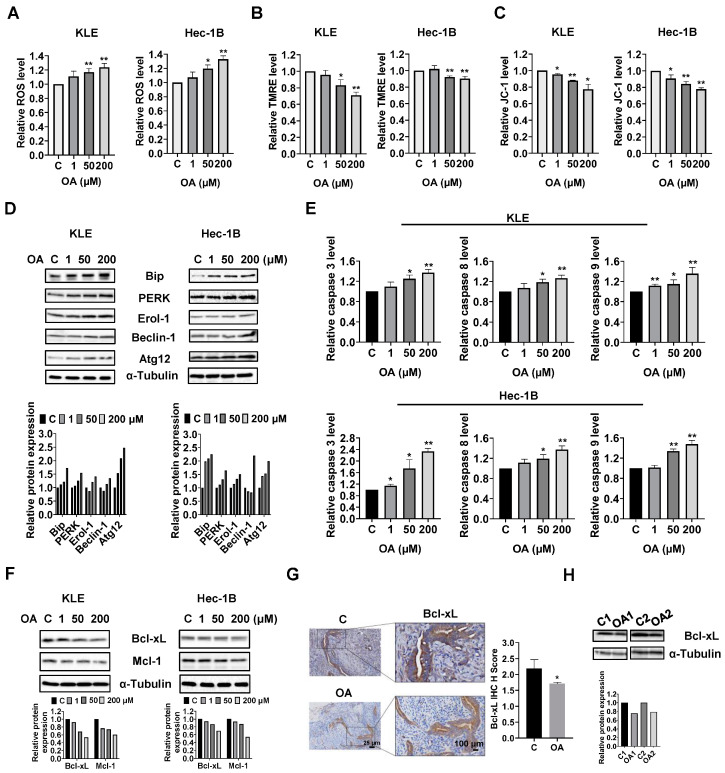
Effects of OA on cellular stress, apoptosis, and autophagy in EC cells and *LKB1^fl/fl^p53^fl/fl^* mice. KLE and Hec-1B cells were treated with 1, 50, and 200 μM OA for 6–14 h. OA significantly increased the ROS levels in both cell lines (**A**). The TMRE assay showed that 50 and 200 μM OA effectively decreased mitochondria membrane potential in KLE and Hec-1B cells (**B**). OA at a dose of 50 and 200 μM resulted in a significant decrease in JC-1 levels in KLE and Hec-1B cells (**C**). Western blotting results revealed that OA increased the expression of BiP, PERK, Erol-1, Beclin-1, and Atg12 proteins after treatment with OA for 24 h (**D**). Cleaved caspase 3, 8, and 9 activities were determined using ELISA. After treatment with 1, 50, and 200 μM OA for 14 h, the activities of cleaved caspase 3, 8, and 9 were increased in both KLE and Hec-1B cell lines (**E**). The expression of Bcl-xL and Mcl-1 was decreased in both cell lines after treatment with different doses of OA for 8 h (**F**). IHC staining showed a decrease in the expression of Bcl-xL in OA-treated *LKB1^fl/fl^p53^fl/fl^* mice compared with that in control mice (**G**). Western blotting results also showed that OA inhibited the expression of Bcl-xL in EC tumor tissues of *LKB1^fl/fl^p53^fl/fl^* mice compared with control mice (**H**). * *p* < 0.05, ** *p* < 0.01.

**Figure 3 cancers-15-05407-f003:**
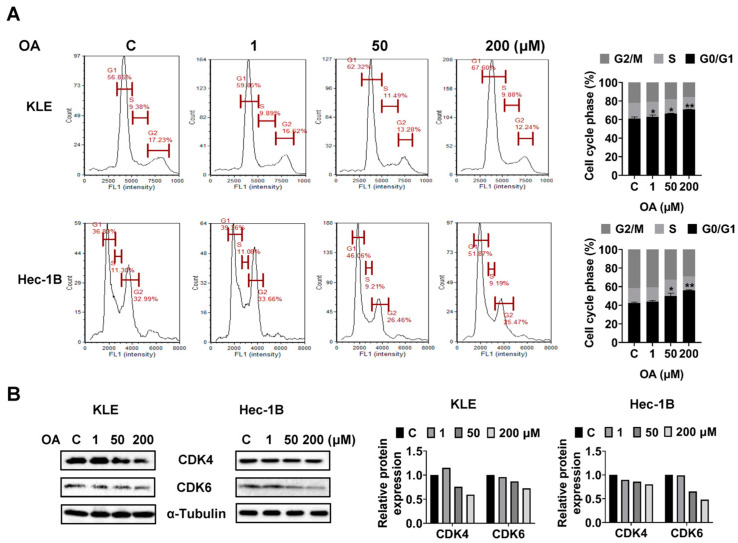
OA induced cell cycle G1 arrest. KLE and Hec-1B cells were treated with 1, 50, and 200 μM OA for 36 h, and cell cycle progression was analyzed using a Cellometer. OA induced cell cycle G1 arrest in both cell lines (**A**). Western blotting was performed to detect the expression of cell cycle-related proteins. OA reduced the expression of CDK4 and CDK6 in both cell lines after OA treatment (**B**). * *p* < 0.05, ** *p* < 0.01.

**Figure 4 cancers-15-05407-f004:**
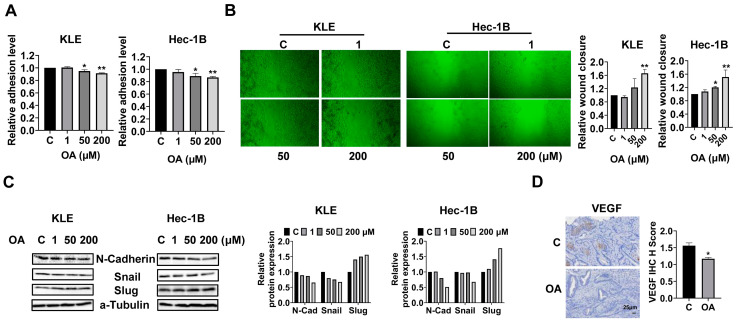
OA inhibited adhesion and invasion. Adhesive ability was detected by laminin-1 assay in the KLE and Hec-1B cells. OA inhibited cell adhesion in the KLE and Hec-1B cells (**A**). The wound healing assay showed that cell migration was inhibited by OA after 28 h of treatment in both cell lines (**B**). Western blotting showed that OA decreased the expression of N-cadherin and Snail and increased the expression of Slug in both cells after 24 h of treatment (**C**). IHC results indicated that treatment with OA for four weeks in *LKB1^fl/fl^p53^fl/fl^* mice inhibited the expression of VEFG in EC tumor tissues compared with that in control mice (**D**). * *p* < 0.05, ** *p* < 0.01.

**Figure 5 cancers-15-05407-f005:**
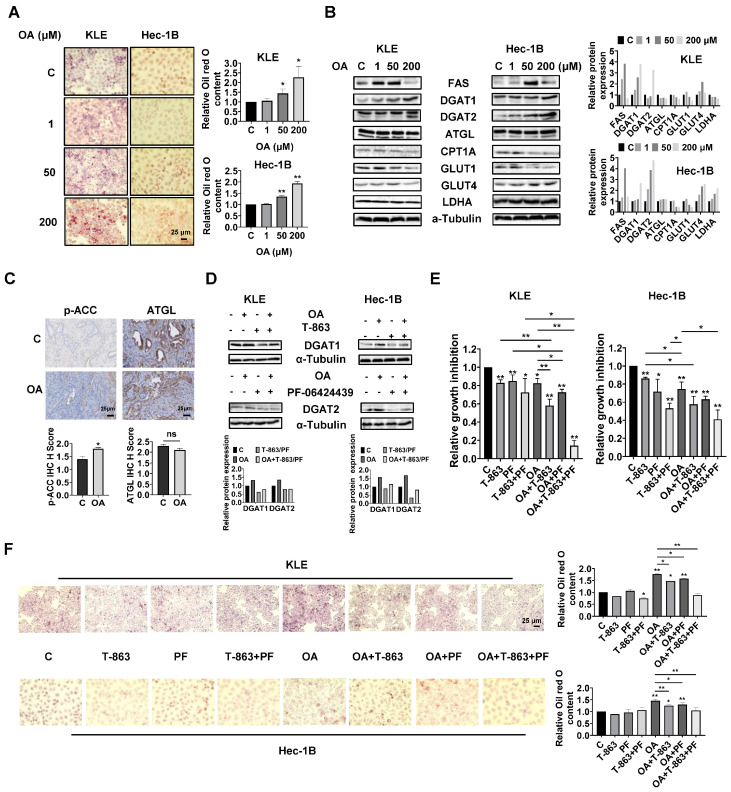
OA induced the accumulation of intracellular lipid droplets in EC cells. The formation of LDs in EC cells was determined by Oil red O staining assay. Treatment of OA at doses of 50 and 200 μM for 24 h significantly increased the content of LDs in both cells (**A**). Western blotting results showed that OA increased the expression of FAS and GLUT4 in both cells, especially at a dose of 50 μM. OA also increased the expression of DGAT1 and DGAT2 and decreased the expression of CPT1A and GLUT1 in both cells after 24 h of treatment in KLE cells. Treating cells with OA for 24 h decreased the expression of LDHA in the KLE cells and increased that in the Hec-1B cells. No significant change in the expression of ATGL in the KLE and Hec-1B cells was detected after treatment of OA for 24 h (**B**). IHC results indicated that treatment with OA for four weeks in *LKB1^fl/fl^p53^fl/fl^* mice increased the expression of phosphorylated-ACC in EC tumor tissues but did not significantly affect the expression of ATGL compared with that in control mice (**C**). Western blotting showed that treatment of T-863 at a dose of 2.5 μM for 24 h effectively inhibited the expression of DGAT1 and that treatment of PF-06424439 at a dose of 5 μM for 24 h decreased the expression of DGAT2 in both EC cells (**D**). MTT assay showed that 2.5 μM T-863 or 5 μM PF-06424439, or the combination of T-863 and PF-06424439, significantly inhibited cell proliferation, and the combination of T-863 or PF-06424439 and 100 μM OA produced more potent inhibition of cell proliferation compared to single agent. The combination of T-863, PF-06424439, and OA exhibited the strongest inhibitory effect on cell proliferation in both cell lines after 72 h of treatment (**E**). Treatment of T-863 (2.5 μM) and PF-06424439 (5 μM) as a single agent showed no significant effects on LD formation in EC cells compared to control group. When treating cells with OA at a dose of 100 μM, either T-863 or PF-06424439 decreased the accumulation of LDs, and the combination of T-863 and PF-06424439 exhibited the strongest inhibitory effect on LD content in both cells (**F**). * *p* < 0.05, ** *p* < 0.01.

**Figure 6 cancers-15-05407-f006:**
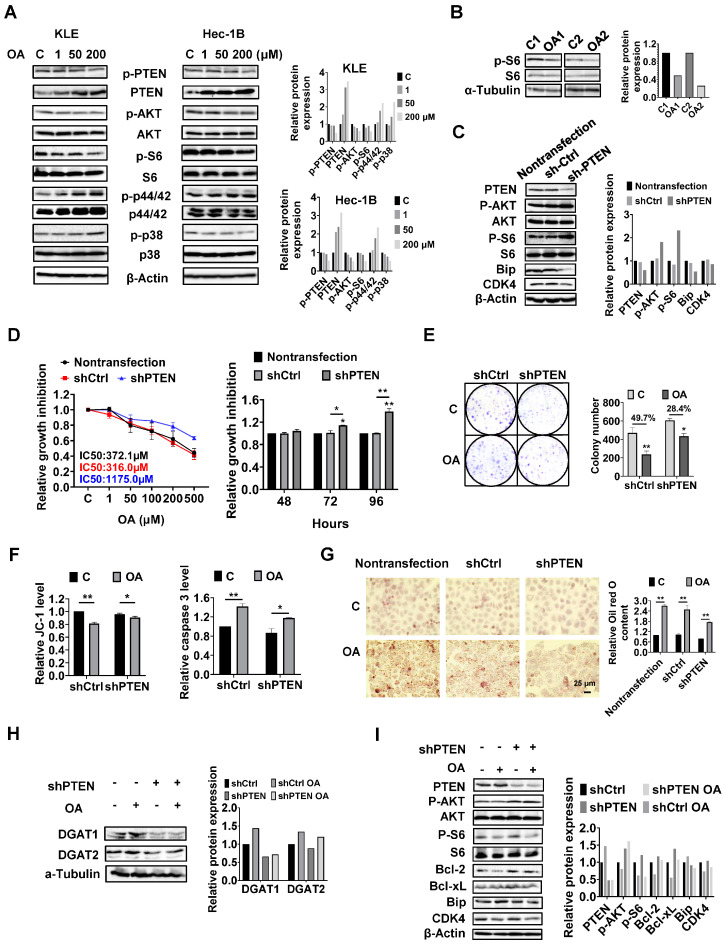
PTEN regulates cell proliferation, stress, apoptosis, and the formation of LDs in EC cells. The KLE and Hec-1B cells were treated with 1, 50, and 200 μM for 24 h. Western blotting results showed that OA significantly increased the expression of PTEN and decreased the expression of phosphorylated-PTEN, phosphorylated-AKT, and phosphorylated-S6 in both cell lines. Treatment of OA also induced the expression of phosphorylated-p42/44 in both cell lines, increased phosphorylation of p38 expression in the KLE cells, and decreased the expression of p38 phosphorylation in the Hec-1B cells (**A**). OA inhibited the expression of phosphorylated S6 in EC tumor tissues from *LKB1^fl/fl^p53^fl/fl^* mice by Western blotting compared to control mice (**B**). Knockdown of PTEN using shRNA significantly decreased the level of PTEN and increased the expression of phosphorylated AKT and phosphorylated-S6 in the Hec-1B cells compared to non-transfected and shCtrl cells. The expressions of Bip and CDK4 were decreased in shPTEN cells compared to non-transfected and shCtrl cells (**C**). Knockdown of PTEN significantly increased the IC50 of OA in the EC cell lines, after 72 h of treatment (**D**). Downregulation of PTEN resulted in increased cell colony numbers compared to shCtrl cells and reduced the inhibitory effect of OA at a dose of 200 μM on cell colony formation compared to shCtrl cells (**E**). Similarly, JC-1 and caspase 3 assays showed that loss of PTEN partially restored mitochondrial membrane potential and cleaved caspase 3 activity in 200 μM OA-treated shPTEN cells (**F**). Treating the non-transfected cells, shCtrl cells, and shPTEN cells with or without OA at a dose of 200 μM for 24 h, the Oil red O results showed that there was no significant difference among the three cells without OA exposure. After treatment of OA (200 μM) for 24 h, relative Oil red O content increased 2.70-fold, 2.48-fold, and 2.21-fold in the non-transfected cells, shCtrl cells, and shPTEN cells, respectively, compared to untreated cells (**G**). Western blotting showed that the expression of DGAT1 and DGAT2 were suppressed after knocking down of PTEN compared to the shCtrl cells, and loss of PTEN also decreased OA-induced expression of DGAT1 and DGAT2 (**H**). The downregulation of PTEN partially reversed the inhibitory effects of OA on the expression of phosphorylated AKT, phosphorylated S6, Bip, Bcl-xL, Bcl-2, and CDK4 (**I**). * *p* < 0.05, ** *p* < 0.01.

**Figure 7 cancers-15-05407-f007:**
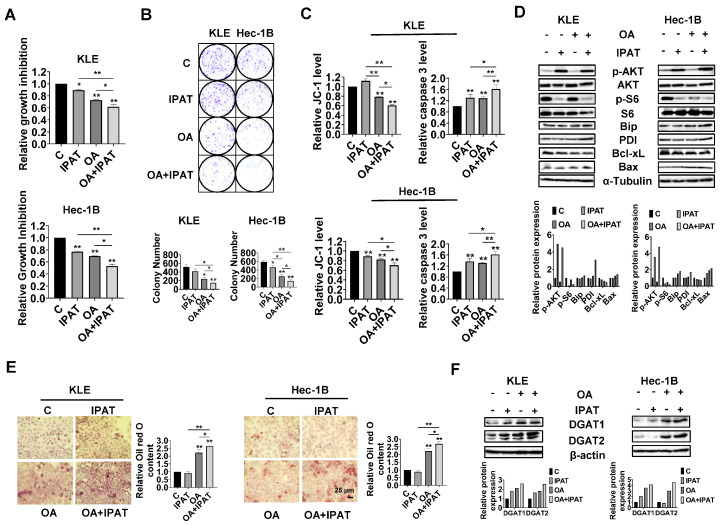
Effect of targeting AKT on OA-induced cell growth and LD formation in EC cells. The KLE and the Hec-1B cells were treated with 20 µM IPAT, 200 μM OA, and the combination of IPAT and OA for 72 h. MTT results showed that the combination treatment produced a more potent inhibitory effect on cell proliferation compared with IPAT or OA alone in both cell lines (**A**). Similar results were obtained in the colony assay. Treatment of OA at a dose of 200 μM in the combination of IPAT at a dose of 20 μM for 48 h exhibited the strongest inhibitory effect on cell colony formation compared with OA or IPAT alone (**B**). The combination of IPAT and OA resulted in the strongest effects on reducing JC-1 levels and increasing cleaved caspase 3 activity in both cell lines compared to IPAT or OA alone (**C**). Western blotting results showed that the combination of IPAT and OA had a stronger inhibitory effect on phosphorylation of S6 compared with IPAT alone, OA alone, and control in both cell lines. Similarly, the combination of IPAT and OA had greater effects on increasing Bax, PDI, and Bip expression and decreasing expression of Bcl-xL than either agent alone (**D**). Results of Oil red O staining showed that treatment of IPAT at a dose of 20 μM for 24 h had no significant effect on the formation of LDs in both cells compared to untreated cells. The combination of OA (200 μM) and IPAT (20 μM) enhanced OA-induced LD formation in both cell lines after 24 h of treatment (**E**). Western blot showed that both OA at a dose of 200 μM and IPAT at a dose of 20 μM increased the expression of DGAT1 in both cells and IPAT showed no significant effect on the expression of DGAT2 in Hec-1B cells after 24 h of treatment. IPAT increased OA-induced expression of DGAT1 and DGAT2 in both cell lines after 24 h of treatment (**F**). * *p* < 0.05, ** *p* < 0.01.

**Figure 8 cancers-15-05407-f008:**
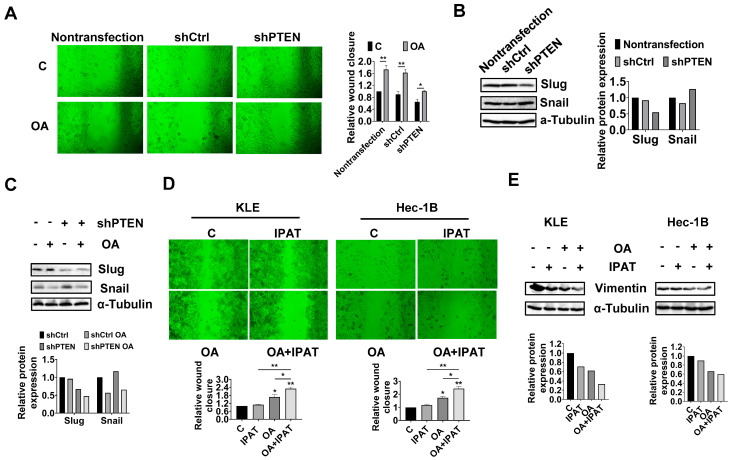
PTEN/AKT/mTOR pathway involved in the effects of OA on adhesion and invasion in EC cells. Wound healing results showed that knockdown of PTEN significantly increased the migration of Hec-1B cells, and the inhibitory effect of OA on cell migration was attenuated after knockdown of PTEN after treatment for 28 h (**A**). Western immunoblotting results showed that knockdown of PTEN decreased the expression of Slug and increased the expression of Snail compared to the non-transfected and shCtrl cells (**B**). Exposure to 200 μM OA for 24 h lead to elevated Slug and decreased Slug in both shCtrl and shPTEN cells, and loss of PTEN attenuated the effects of OA on the expression of Snail and Slug compared to shCtrl cells (**C**). The results of wound healing assay showed that 20 μM IPAT alone did not affect the migratory ability of either cell line. However, the combination of IPAT and OA significantly increased the inhibitory effects of OA on cell migration in both cell lines compared to treatment with IPAT or OA alone after 28 h of treatment (**D**). Western immunoblotting results showed that treatment of KLE and Hec-1B cells with 200 μM OA reduced the expression of Vimentin compared to untreated cells. The combination of IPAT and OA showed a more potent inhibitory effect on Vimentin in both cells (**E**). * *p* < 0.05, ** *p* < 0.01.

## Data Availability

All data generated or analyzed during this study are included in this article. The datasets used and/or analyzed during the current study are available from the corresponding authors upon reasonable request.

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
