# Peer review of "Oleic Acid Exhibits Anti-Proliferative and Anti-Invasive Activities via the PTEN/AKT/mTOR Pathway in Endometrial Cancer"

_cancers, 2023, doi:10.3390/cancers15225407_

Round 1
Reviewer 1 Report
Comments and Suggestions for Authors
You maybe considering of carrying out additional research and works in the following areas to strengthen the results of the study "Oleic acid demonstrates anti-proliferative and anti-invasive actions via the PTEN/AKT/mTOR pathway in endometrial cancer."
Comments-
1-Kindly elaborate why you only chose oleic acid and not other long-chain fatty acids.
2-Look into how oleic acid affects the PTEN/AKT/mTOR pathway's downstream targets. Examine in considerable detail how oleic acid affects the important regulators involved in cell cycle control and apoptosis.
3-To acquire a more thorough understanding of the dose-response relationship, particularly for the cell lines with high IC50 values, examine a wider range of oleic acid concentrations. This may offer information about the ideal dosage for therapeutic uses.
4-The author has excluded statistical analysis from numerous Western blotting densitometry analysis figures.
5-Many images, including Figure 8 A&D, are of low quality.
6-The Author needs to pay close attention to the figure legends where they have either failed to include or missed to describe the image's data. (Like Figure 5F, and Figure 6G)
Comments on the Quality of English LanguageHere, the authors also require attention.
Author Response
1-Kindly elaborate why you only chose oleic acid and not other long-chain fatty acids.
Obesity is the most important risk factor for carcinogenesis and progression of endometrial cancer. Fatty acid metabolism seems to have dual functions in the regulation of the development of cancer in preclinical models of cancer. However, studies on the association between fatty acid metabolic profiles and endometrial cancer are limited, and they have conflicting results in epidemiological and clinical data. A recent study demonstrated the downregulation of cholesterol, arachidonic acid, palmitic acid, oleic acid, stearic acid, and linoleic acid in the serum of patients with endometrial cancer, suggesting that these metabolites may be most relevant to endometrial cancer. Additionally, oleic acid is the most abundant fatty acid in human adipose tissue, and second in abundance in human tissues overall, following palmitic acid. Oleic acid is also the most common monounsaturated fatty acid in nature. Therefore, we believe that understanding the effects of oleic acid on endometrial cancer tumor growth will greatly help us design dietary interventions in future clinical trials for endometrial cancer.
2-Look into how oleic acid affects the PTEN/AKT/mTOR pathway's downstream targets. Examine in considerable detail how oleic acid affects the important regulators involved in cell cycle control and apoptosis.
Thanks for your comments.
3-To acquire a more thorough understanding of the dose-response relationship, particularly for the cell lines with high IC50 values, examine a wider range of oleic acid concentrations. This may offer information about the ideal dosage for therapeutic uses.
Thanks for your comments. This is a perfect point. In the initial experiment, we selected the dose of OA from 0.001 to 2000 μM based on the concentration of OA in the plasma of normal adults to investigate the effect of oleic acid on cell proliferation after 72 hours of treatment. We found that oleic acid at a dose of 1 uM started to inhibit cell proliferation and totally inhibited cell proliferation at the doses of 750 to 1000 uM. Therefore, we presented the inhibitory effect of 0.1 to 500 uM OA doses on cell proliferation in the MTT results (Fig 1A). In subsequent experiments, we selected three doses (1, 50, and 200 uM) of OA to study the effect of OA on growth inhibition, because many research studies showed that the physiological concentration of oleic acid is generally below 500 μM.
4-The author has excluded statistical analysis from numerous Western blotting densitometry analysis figures.
The authors appreciated the reviewer’s comments. Data from western blotting densitometric analysis represented only the protein expression results shown in the Figures.
5-Many images, including Figure 8 A&D, are of low quality.
Thanks for your comments. We have re-uploaded these pictures in high quality.
6-The Author needs to pay close attention to the figure legends where they have either failed to include or missed to describe the image's data. (Like Figure 5F, and Figure 6G).
We are sorry for the mistakes. We have corrected the mistakes in the figure legends.
Reviewer 2 Report
Comments and Suggestions for Authors
The authors described that oleic acid exhibits anti-proliferative and anti-invasive activities via the PTEN/AKT/mTOR pathway in endometrial cancer. Remarkably, the LKB1fl/flp53fl/fl mouse model of EC shows that OA promotes lipogenesis and LD formation while suppressing cell proliferation, invasion, and tumor progression. Based on this study, OA decreases EC cell proliferation while raising functional PTEN. However, the study is interesting, and the authors should address the following comments:
1. Figure 4c demonstrates that OA-induced slug expression Most polyphenolic substances decrease slug expression. In this study, OA decreased the expression of snail while increasing the expression of slugs. Describe.
2. The authors need to specify whether tryptophan or pAKT serine have been used in this investigation.
3. The expression of pAKT should be reduced when OA and ShPTEN are combined. Figure 6I of the study shows enhanced pAKT expression. Describe.
4. The expression of pAKT should be reduced when OA and IPAT are combined. Figure 7D of the study shows enhanced pAKT expression. Describe.
5. Figure 8B: Slug expression should rise with shPTEN knockdown. Expression was reduced in the study. Describe.
Author Response
Reviewer 2
The authors described that oleic acid exhibits anti-proliferative and anti-invasive activities via the PTEN/AKT/mTOR pathway in endometrial cancer. Remarkably, the LKB1fl/flp53fl/fl mouse model of EC shows that OA promotes lipogenesis and LD formation while suppressing cell proliferation, invasion, and tumor progression. Based on this study, OA decreases EC cell proliferation while raising functional PTEN. However, the study is interesting, and the authors should address the following comments:
- Figure 4c demonstrates that OA-induced slug expression Most polyphenolic substances decrease slug expression. In this study, OA decreased the expression of snail while increasing the expression of slugs. Describe.
The authors appreciate the reviewers’ comments. Slug and Snail are aberrantly expressed and regulate many kinds of fundamental processes, including cell proliferation, apoptosis, and cell motility in cancers, including endometrial cancer. Recent studies demonstrated a complex relationship between slug and snail in cancer cells. For example, reciprocal expression levels of snail and slug exist in different breast cancer cells. The knockdown of slug appears to upregulate snail expression, and snail siRNA also appears to upregulate Slug expression in oral cancer cells. No existing theory can explain this phenomenon. In our study, we found that OA inhibited the expression of snail and increased the expression of slug in KLE and HEC-1A cell lines. However, our recent study found that Ipatasertib (IPAT, an AKT inhibitor) can inhibit the expression of slug and snail in serous endometrial cancer, suggesting that different agents with different mechanisms exhibit different effects on slugs and snails.
- The authors need to specify whether tryptophan or pAKT serine have been used in this investigation.
In this study, we used a pAKT ser473 antibody. We have added this information to the section of Results.
- 3. The expression of pAKT should be reduced when OA and ShPTEN are combined. Figure 6I of the study shows enhanced pAKT expression. Describe.
Answer: Thanks for your comments. When knocking down PTEN in Hec-1B cells, the inhibitory effects of OA were suppressed and not significant. There may not be a difference between the two groups.
- The expression of pAKT should be reduced when OA and IPAT are combined. Figure 7D of the study shows enhanced pAKT expression. Describe.
Ipatasertib (IPAT), is a highly selective inhibitor of phosphorylated AKT which competitively binds and inactivates the p-AKT complex and disrupts the mTOR pathway. Because it competes with ATP for binding sites, it prevents dephosphorylation of the p-AKT complex in a dose-dependent manner (PMID: 23287563). Similar results can be found in previous publications from our laboratory (PMID: 35812065; PMID: 36773034).
- Figure 8B: Slug expression should rise with shPTEN knockdown. Expression was reduced in the study. Describe.
We appreciated the reviewer’s comments. Please refer to answer 1.
Reviewer 3 Report
Comments and Suggestions for Authors
In this manuscript, the authors demonstrated that oleic acid inhibits endometrial cancer via PTEN/AKT/mTOR signaling pathway. The authors provided solid results.
Comments:
In Fig. 3B, 4C, 5B, 6A, 6B, 6C, 6H, 6I, 7D, 8B, 8C, 8E, the quantification should provide standard deviation on the bar.
The authors should provide a graphic model that would be favorable for readers to understand this paper.
All Figures should be enlarged, several results are too small and hard to read.
Author Response
Reviewer 3
In this manuscript, the authors demonstrated that oleic acid inhibits endometrial cancer via the PTEN/AKT/mTOR signaling pathway. The authors provided solid results.
Comments:
In Fig. 3B, 4C, 5B, 6A, 6B, 6C, 6H, 6I, 7D, 8B, 8C, 8E, the quantification should provide standard deviation on the bar.
The authors appreciated the reviewer’s comments. The protein densities shown in the figures represent only the Western blot results in the figures.
The authors should provide a graphic model that would be favorable for readers to understand this paper.
We have created a graphic abstract following the reviewer’s suggestions.
All Figures should be enlarged, several results are too small and hard to read.
Thanks for the comments. We have tried our best to adjust each figure.
Round 2
Reviewer 1 Report
Comments and Suggestions for Authors
Despite the authors' efforts to modify the article, it is worth noting that the quality of the migration assay images shown in Figures 4B and 8A and B could have been further improved. In order to enhance the efficiency of the r, it is advised to improve these images.
Comments on the Quality of English Language
no any
Author Response
Despite the authors' efforts to modify the article, it is worth noting that the quality of the migration assay images shown in Figures 4B and 8A and D could have been further improved. In order to enhance the efficiency of the r, it is advised to improve these images.
The authors thank the reviewers for their comments. Due to the default settings of microscope cameras, it is difficult to increase the number of pixels in the images. We try our best to adjust size, contrast, and brightness to improve the clarity of the images.
Reviewer 2 Report
Comments and Suggestions for Authors
The authors appropriately responded to every comment.
Author Response
The authors thank the reviewers for their efforts in improving our paper
Reviewer 3 Report
Comments and Suggestions for Authors
The manuscript is well-revised.
Author Response
The authors thank the reviewers for their efforts in improving our paper.